# Extraction and Characterization of Antioxidant Peptides from Fruit Residues

**DOI:** 10.3390/foods9081018

**Published:** 2020-07-29

**Authors:** Saúl Olivares-Galván, María Luisa Marina, María Concepción García

**Affiliations:** Departamento de Química Analítica, Química Física e Ingeniería Química, Instituto de Investigación Química “Andrés M. del Río”, Universidad de Alcalá, Ctra. Madrid-Barcelona, km. 33.600, E-28871 Alcalá de Henares, Madrid, Spain; saul.olivares@uah.es (S.O.-G.); mluisa.marina@uah.es (M.L.M.)

**Keywords:** fruit residues, antioxidant, peptides, extraction

## Abstract

Fruit residues with high protein contents are generated during the processing of some fruits. These sustainable sources of proteins are usually discarded and, in all cases, underused. In addition to proteins, these residues can also be sources of peptides with protective effects against oxidative damage. The revalorization of these residues, as sources of antioxidant peptides, requires the development of suitable methodologies for their extraction and the application of analytical techniques for their characterization. The exploitation of these residues involves two main steps: the extraction and purification of proteins and their hydrolysis to release peptides. The extraction of proteins is mainly carried out under alkaline conditions and, in some cases, denaturing reagents are also employed to improve protein solubilization. Alternatively, more sustainable strategies based on the use of high-intensity focused ultrasounds, microwaves, pressurized liquids, electric fields, or discharges, as well as deep eutectic solvents, are being implemented for the extraction of proteins. The scarce selectivity of these extraction methods usually makes the subsequent purification of proteins necessary. The purification of proteins based on their precipitation or the use of ultrafiltration has been the usual procedure, but new strategies based on nanomaterials are also being explored. The release of potential antioxidant peptides from proteins is the next step. Microbial fermentation and, especially, digestion with enzymes such as Alcalase, thermolysin, or flavourzyme have been the most common. Released peptides are next characterized by the evaluation of their antioxidant properties and the application of proteomic tools to identify their sequences.

## 1. Introduction

The growing world population, together with the increasing popular awareness about healthy nutritional habits, has promoted a massive rise in fruit production [1,2,3]. This trend has boosted the release of fruit residues. A production residue is defined as a material that is not deliberately produced in a productive process. If that residue has a certain use, is ready for use without further processing, and has to be produced as an integral part of the production process, then that residue is called “by-product.” If any of those three conditions are not met, the residue is called “waste” [4].

It has been estimated that approximately 50% of the original weight of fruits becomes waste in the form of peels, pomaces, seeds, and unripe or damaged fruits, which is an unsustainable rate [5]. The usual strategies for the management of these wastes include landfilling and incineration. These practices demand high amounts of oxygen, are associated with the emission of greenhouse gases, and create a platform for pathogenic bacteria and pests. Furthermore, unacceptable odors are generated during their biodegradation [6,7]. Nevertheless, many of these wastes can be reused, and some are currently employed in composting and animal feeding. A more efficient use for these resources is as feedstock in biorefineries, replacing petrochemical-based matter to produce high value-added products such as chemicals, materials, and fuels [8].

Additionally, many research works have shown that fruit byproducts contain large amounts of many phytochemicals and essential nutrients. Pectin, polyphenols, carotenoids, flavonoids, and fiber are some of the functional and nutritional constituents of fruit residues that have attracted the greatest interest [9]. However, bioactive proteins and peptides, which have been more explored in foods from animal origin, are also present in plants [10]. Fruit seeds, for example, are usually the main constituents of fruit residues and store large quantities of proteins and peptides, as well as lipids and carbohydrates, which constitute the plant’s food reserves in its first stages of growth [11].

## 2. Antioxidant Peptides in Fruit Residues

Bioactive peptides can be defined as food components with a positive effect on body functions or conditions beyond their nutritional effects and that may ultimately influence health in a positive manner [12]. A bioactive peptide usually contains 2–20 amino acid residues and can exhibit different biological functions depending on its chemical structure, length, and amino acid composition [13]. Regardless of the kind of sample, peptides can exist as an independent entity or, more often, in a latent state as part of the protein sequence. In this case, the release of peptides requires the hydrolysis of parent proteins [12,14].

Bioactive peptides can be added to foods to improve their functionality (functional foods) or can be used in the manufacture of nutraceuticals. Peptides have shown different bioactivities, including antimicrobial, anticancer, antiviral, hemolytic, and antihypertensive activity, among others, with antioxidant activity being one of the most researched [15].

Oxidative stress is caused by the presence of high amounts of reactive oxygen species (ROS) that overcome endogenous antioxidant defense mechanisms. Maintained oxidative stress can lead to the development of serious diseases by damaging important biomolecules such as DNA, proteins, and lipids [16]. Different studies have demonstrated that antioxidant intake is inversely related to cellular death, aging, and the development of diseases such as diabetes and cancer [17].

In addition to the health benefits, antioxidant peptides can be added to food systems to reduce oxidative changes during storage [18]. While lipid peroxidation inhibition is the most important mechanism, peptides are also capable of reducing the oxidative modification of intact proteins. Vegetable protein hydrolysates are already allowed to be used as food additives in the United States. Moreover, antioxidant peptides could also be added to cosmeceutical products to neutralize free radicals, thus preventing the signs of aging skin [19].

There are some common features within antioxidant peptides. They usually present a large amount of hydrophobic amino acids, such as leucine, alanine, and phenylalanine, that enhance hydrogen-transfer and lipid peroxyl radical trapping, promote their accessibility to hydrophobic targets, and make it easier to pass through cell membranes [12,16,20]. On the other hand, the presence of aromatic amino acids, such as histidine, tyrosine, and tryptophan within a peptide sequence has also been found to be related to antioxidant properties due to their capability to donate electrons to free radicals, thus converting them into stable molecules [20,21]. However, aromatic and hydrophobic amino acids impart a bitter taste to protein hydrolysates, which may create organoleptic problems when used as food additives [18].

Molecular weights between 0.5 and 1.5 kDa are also a common feature within antioxidant peptides [22]. Antioxidant peptides have been obtained from rapeseed residues (*Brassica napus*) [23,24], peels of pomegranate (*Punica granatum*) [3,25] and mango (*Mangifera indica*) [26], and seeds of apricot (*Prunus armeniaca*) [11,27], peach (*Prunus persica L*.) [11,16,28,29], bottle gourd (*Lagenaria sciceraria*) [30,31,32], cherry (*Prunus cerasus L.*) [11,33], olive (*Olea europaea*) [11,16,17,34], plum *(Prunus domestica L.*) [11,14,29,35], tomato (*Solanum lycopersicum*) [21,36,37,38,39,40,41,42,43], wax gourd (*Benincasa hispida*) [44], jujube (*Ziziphus jujube*) [45,46], muskmelon (*Cucumis melo*) [47], watermelon (*Citrullus lanatus*) [32,48,49,50,51,52,53], papaya (*Carica papaya*) [54], Chinese cherry (*Prunus pseudocerasus*) [55], African breadfruit (*Treculia Africana*) [56], pumpkin (*Cuburbita pepo* [57,58,59,60,61,62] and *Cucurbita moschata* [20,32,63,64]), and date (*Phoenix dactylifera*) [65,66]. Some of these residues are released in the extraction of the oil fraction of seeds (such as pumpkin seeds or rapeseed) and of fruits (such as olives) or during the processing of fruits or vegetables such as pomegranate, *Prunus* fruits, tomato, muskmelon, watermelon, papaya, and mango.

## 3. Obtaining Antioxidant Proteins and Peptides from Fruit Residues

The exploitation of sustainable sources of proteins requires the development of suitable methodologies. A general procedure followed for obtaining proteins and peptides with antioxidant properties from fruit residues is shown in Figure 1. Usually, the fruit residue is dried and ground before the extraction in order to avoid microbiological contamination during storage and to promote the penetration of the extracting solvent into the solid matrix. Furthermore, when the lipid content in the fruit residue is high and disturbs the extraction of proteins, it is necessary to include a previous defatting step. Typically, a preliminary extraction with hexane is the selected procedure. Sometimes, there is also a sieving step before the extraction to obtain a more homogeneous material [21,43,45,46,47]. Then, the extraction of proteins is performed.

Conventional and non-conventional techniques have been tested for the extraction of proteins from fruit residues. Protein extraction from plant tissues is currently carried out by strategies that involve the use of polluting reagents and volatile organic solvents, and they result in very low yields. Moreover, some of these reagents are not food-grade and cannot be employed in industrial applications. Protein extraction from sustainable sources, such as fruit residues, urges the development of alternative strategies with a lower environmental impact and a higher protein yield. The application of sustainable techniques that require less polluting reagents and less energy are of special interest. Another important aspect in protein extraction is selectivity. The usual lack of the selectivity of extraction procedures makes, in many cases, an additional step to purify proteins necessary.

Extracted proteins are usually submitted to a hydrolysis process to obtain peptides. This step generally requires the use of food grade enzymes. Extracted proteins and peptides are sometimes fractionated based on different parameters—mainly molecular mass, solubility, and hydrophobicity—to have a deeper inside on properties of proteins/peptides within different fractions.

The evaluation of antioxidant properties in extracts, hydrolysates, and/or fractions involves the use of different in vitro assays based on different mechanisms. Additional studies evaluating the capacity of proteins and peptides to reduce the oxidative stress on cells cultures and animal models are very interesting to confirm in vitro results. In some cases, a further characterization of extracts is carried out by the identification of peptides using tandem mass spectrometry.

## 4. Techniques Used in the Extraction and Purification of Proteins

Protein extraction requires the breakdown of tissues, cell membranes, and cell walls in order to release intracellular material. The difficulty is high in the case of plant tissues due to the presence of large vacuoles, the rigidity and thickness of cell walls, and the heterogeneity of proteins. Moreover, the presence of lipids, polysaccharides, or phenolic compounds can interfere with the extraction of proteins [14].

The amount and characteristics of proteins in fruit residues is highly variable being not possible to generalize. Indeed, the dried peels of pawpaw, pineapple, mango, apple, banana, orange, pomegranate, and watermelon present between 2.8%, for apples, and 18.1%, for pawpaw, of crude proteins [67]. On the other hand, fruit seeds, in general, show a higher protein content. For example, the seeds of cherry, pumpkin, papaya, watermelon, mango, jackfruit, orange, melon, peach, and Surinam cherry present a protein content ranging from 6%, for the mango seeds, to 39%, for the pumpkin seeds [68,69]. Some functional proteins have been found in some residues, such as passiflin, a dimeric protein from passion fruit seeds that exhibits antifungal and anticancer activities [70].

The performance and sustainability of protein extraction can be improved by favoring physical contact between the extracting medium and proteins and by using more environmentally friendly solvents. Physical contact between the extracting medium and target compounds can be promoted by the use of ultrasound-assisted extraction, pressurized liquid extraction, microwave-assisted extraction, or by the application of electric energy (pulsed electric fields or high-voltage electrical discharges). These techniques are being implemented in the extraction of proteins from fruit residues at the laboratory scale. Additionally, the introduction of nanomaterials in the extraction and purification of proteins and the use of deep eutectic solvents are promising approaches to increase the sustainability of extraction procedures. Table 1 summarizes the methods employed to extract proteins and peptides from fruit residues.

### 4.1. Solid–Liquid Conventional Extraction

Traditional methods for the extraction of proteins employ aqueous buffers that can contain reducing and chaotropic reagents (dithiothreitol, mercaptoethanol, urea, etc.), surfactants, etc. Moreover, protein extraction is usually followed by a final purification step that involves the use of an organic solvent or the acidification of the sample.

Since most food proteins have low isoelectric points, the extraction of proteins at pHs ranging from 7.5 to 12 using NaOH solutions and stirring, followed by their acidic precipitation at pHs from 3.8 to 5.3 is very popular. Incubation temperatures of 40–50 °C have been employed to promote protein solubilization. This strategy has been used in the extraction of proteins from the seeds of tomato, jujube, watermelon, Chinese cherry, African breadfruit, and pumpkin, as well as in pumpkin oil cake [21,37,41,42,43,45,48,50,52,55,56,57,58,59,60,61,62]. Despite the popularity and industrial applicability of this procedure, its selectivity and extraction yield, in general, are very low. Extraction yield has been found to range from 23%, in the case of the African breadfruit seeds, to 56%, in the case of the pumpkin seeds [56,59]. After a purification step by acid precipitation, the protein contents of isolates reached 80%, 82%, and 90% for tomato, Chinese cherry, and African breadfruit seeds, respectively [42,55,56]. In addition to the acid precipitation of proteins, other purification protocols using acetone or (NH₄)₂SO₄ have been employed in the case of the jujube seeds [45,46]. The main advantage of using (NH₄)₂SO₄ is its non-denaturing character. Additionally, alkaline extraction results in protein degradation, the reduction of protein solubility at neutral pH, and poor technological functionalities.

Alternatively, proteins have been extracted at other pHs. Parniakov et al. [26,54] used different pHs (2.5, 6.0, and 11.0) for extracting proteins from mango peels and papaya seeds, but, again, they observed the highest extraction at pH 11.0. Phosphate and Tris(hydroxymethyl)aminomethane (Tris)-HCl buffers were used in the extraction of milled rapeseed, bottle gourd, pumpkin (*Cucurbita moschata*), watermelon, wax gourd, and jujube seeds [23,30,32,44,45]. Under the same extraction conditions, pumpkin (*Cucurbita moschata*), watermelon, and bottle gourd seeds yielded isolates with 46%, 39%, and 49% protein contents, respectively. In some cases, buffers contained additives to avoid protease activity (ethylenediaminetetraacetic acid; EDTA) or to promote protein solubilization. Indeed, proteins are folded and usually form insoluble aggregates that constitute a limitation for their extraction. Dithiothreitol (DTT) is a usual additive that reduces disulfide bonds between cysteine residues and, thus, improves the extraction of proteins. Urea, for example, is a chaotropic agent that is added to disrupt hydrogen bonds between amino acids. Surfactants, such as SDS and Triton X-100 have also been added to the extracting media at low concentrations (below the critical micelle concentration). SDS is a denaturing surfactant that disrupts cell membranes and breaks interactions within proteins. Triton X-100 is a non-denaturing surfactant that cannot penetrate into proteins and disrupt interactions, but it can associate with hydrophobic parts of the protein to promote solubilization [23,30,44]. While extractions in an alkalized medium is an usual procedure in the food industrial environment, the use of a Tris-HCl buffer is not suitable. Moreover, additives such us DTT, SDS, and urea are not food-grade reagents and cannot be used in the manufacture of products for animal or human consumption.

The Osborne method [71] has also been employed for the extraction and fractionation of proteins based on their solubility in different media: water (albumins), salt solution (globulins), alkaline solution (glutelins), and alcoholic solution (prolamins). This methodology was applied for the extraction of proteins from pumpkin (*Cucurbita moschata)*, bottle gourd, muskmelon, and watermelon seeds [20,31,47,49]. In all cases, seeds were previously defatted obtaining powders with 56–69% proteins. Most proteins were extracted using salt or alkaline solutions, while lower proteins were extracted using alcoholic solutions. Pumpkin seeds, for example, resulted in a globulin fraction that held the highest protein content (46%), followed by the glutelin fraction (39%), the albumin fraction (23%), and the prolamin fraction (12%). The trend of showing a high protein globulin fraction and a low protein prolamin fraction was common with other seeds such as the watermelon seeds and bottle gourd seeds. Unlike them, melon seeds showed the highest protein content in the glutelin fraction (81%), although the prolamin (6%) fraction was, again, the fraction with the lowest protein content. 

### 4.2. Ultrasound-Assisted Extraction of Proteins

Extraction using high-intensity focused ultrasound (HIFU), first developed around 1950, has been widely employed for the acceleration of these procedures. HIFU provides mechanical energy in the form of acoustic energy and extraction is based on a phenomenon known as cavitation. Ultrasonic waves generate rapid changes in pressure within a solution that lead to the formation of small gas bubbles that collapse and thereby release a high amount of energy. This energy promotes the breakdown of tissues and cell walls, followed by the extraction of proteins [72].

The HIFU extraction of proteins has often been carried out with a Tris-HCl buffer that could also contain SDS and DTT at low concentrations (0.5–1% and 0.1–0.5%, respectively). The optimization of this process required tuning the extraction time, the ultrasound amplitude, the concentration of SDS and DTT, and the sample:solvent ratio. Under optimal conditions, HIFU has been employed for the extraction of proteins from *Prunus* fruits (plum, peach, cherry, and apricot), olive seeds [11,14,16,17,29,33,35], and pomegranate peels [3]. An ultrasound amplitude of 30% has enabled the reduction of extraction times from hours to 1–5 min. 

However, despite being an old technique, upscaling to pilot or industrial use has failed to succeed to date [73], and the solvent and additives used here are not suitable for food production.

### 4.3. Pressurized Liquid Extraction

Pressurized liquid extraction (PLE) uses temperatures and pressures in the ranges of 50–200 °C and 35–200 bar, respectively, to extract desired compounds. The high temperature enhances the solubility and mass transfer rate while reducing the viscosity and surface tension of solvents. The high pressure allows the solvent to rise above the normal boiling point temperature while keeping their liquid state. These conditions favor the penetration of solvents into the sample matrix and the analyte mass transfer. PLE requires a much lower amount of solvents than conventional solid–liquid extraction, and, in addition, it results in a higher yield and a reduced extraction time. Furthermore, PLE is usually carried out using environmentally friendly solvents such as water or ethanol [78]. Ethanol can be produced from raw agricultural materials (cereals and sugar beet) and from waste and residues (straws). Though PLE has been mostly applied to the extraction of small molecules, it can also be useful in protein extraction, as shown in previous works devoted to the extraction of algae proteins [79,80].

In the case of fruit residues, this technique has only been used for the extraction of proteins from pomegranate peels [25]. In order to achieve the highest extraction yield, different parameters were optimized: extracting the solvent, temperature, static time, and the presence of additives in the solvent. The highest extraction yield was obtained when using 70% (v/v) ethanol as the extracting solvent and a high temperature (120 °C). No significant differences were observed by increasing the extraction time over 3 min, repeating cycles, or adding DTT or urea to the extracting solvent. Under optimized conditions, the extraction yield (9 mg of proteins/g of pomegranate peel) was lower than that obtained using HIFU [3] (15 mg/g). Further studies enabled researchers to observe that the proteins extracted with every technique were different and that both techniques could be complementary to obtain a more comprehensive extraction of proteins from pomegranate peels [25].

Moreover, the use of non-toxic, cheap, and environmentally friendly solvents in small quantities, as well as its easy automation, make this technique a great candidate for food industry applications [73]. Nevertheless, working with ethanol at the industrial scale requires certain safety conditions.

### 4.4. Extraction Using Deep Eutectic Solvents

Deep eutectic solvents (DES) are sustainable extractants that are usually derived from renewable resources [81]. DES are obtained by mixing two solid organic compounds, a hydrogen-bond acceptor (HBA), such as quaternary ammonium salts, and a hydrogen-bond donor (HBD), such as amides, alcohols, and acids, at an appropriate molar ratio (eutectic composition). HBD and HBA associate with each other by means of hydrogen bond interactions [25]. DES formation is based on a phenomenon called freezing point depression. The result of that mixing is a liquid solvent at relatively low temperatures. Choline chloride is the most used HBA, while urea, citric acid, glucose, tartaric acid, succinic acid, and glycerol are other usual HBDs employed in the synthesis of DES [81,82,83]. DES have been mostly used in the extraction of small compounds, although some DES have been applied in the extraction of some standard proteins (bovine serum albumin, papain, and wheat gluten) [54,55] and in the extraction of proteins from brewers spent grain [84].

More recently, DES have been employed for the extraction of proteins from pomegranate peels [25]. Different DES were synthesized, and their extraction capacities were compared. Choline chloride and sodium acetate were employed as HBAs, while different HBDs were tried (ethylene glycol, glycerol, acetic acid, glucose, sorbitol, and acetic acid). A choline chloride:acetic acid DES was selected as the most effective. The acceleration of extraction was possible using HIFU at 60% for 11 min. Under these conditions, DES extracted 20 mg of proteins/g of pomegranate peel. A comparison with results obtained for the same sample with HIFU and PLE showed that DES had a higher protein extraction capability (HIFU (15 mg of proteins/g of pomegranate peel) [3] and PLE (9 mg proteins/g of pomegranate peel) [25].

While DES seem to comprise a promising group of green solvents, further research is needed in order to confirm whether they are nontoxic to animals and the environment [85]. Moreover, its applicability at the industrial scale is, so far, compromised by the high price of DES. Further studies on the possible reutilization of DES are very interesting in this regard.

### 4.5. Microwave-Assisted Extraction

Microwave-assisted extraction (MAE) uses microwave radiation to favor extraction processes. The overall effect is the warming of the sample due of the dissipation of the radiation energy by thermal conduction, in the case of ions, or by the rotation of dielectric dipoles and friction with the solvent, in the case of polar molecules. MAE has been widely employed for the extraction of small molecules and, to a lesser extent, for the extraction of proteins [86,87].

The extraction of proteins from pumpkin (*Cucurbita moschata*) seeds was performed using MAE and different DES as extracting solvents [64]. Poly (ethylene glycol) (PEG 200) was employed as the HBD, while several HBAs were tried. A PEG 200:choline chloride mixture at a 3:1 ratio was selected. The performance of MAE was compared with results obtained by using a conventional water bath extraction, the extraction using a HIFU probe, and the extraction using both MAE and HIFU, using the same DES as the extracting medium. Ultrasound–microwave synergistic extraction showed a better average extraction efficiency (94%) with less solvent and shorter extraction times (4 min) than water bath extraction (1 h), MAE (6 min), and HIFU (30 min).

Though this technique can be scaled-up, its application is complex due to the non-homogenous depth of microwave radiation that results in a non-uniform sample heating. Moreover, it is necessary to work under proper conditions to avoid potential hazards.

### 4.6. Extraction Using Pulsed Electric Field (PEF) and High Voltage Electrical Discharge (HVED)

The application of electric discharges has been used in the food industry to increase food shelf life since it has the ability to inactivate microorganisms. The technique is based on the application of electric discharges that create currents and bubbles (cavitation bubbles) that expand and implode, thus causing pressure variations of up to 100 bar. These pressure changes allow for the permeation of cell walls in a controlled way, unlike classical treatments in which tissue structure is entirely disrupted, thus losing its selectivity and becoming permeate to all intracellular compounds. This technique has a high potential in the extraction of proteins, although it has been scarcely used for this purpose. Moreover, it allows for work at low temperature conditions, which can also be an advantage over other techniques that require the use of high temperatures. Indeed, non-thermal emerging techniques have been proposed to shorten the processing time, to increase recovery yield, to control Maillard reactions, to improve product quality, and to enhance extract functionality [34,88].

Extraction using pulsed electrical field (PEF) is a non-thermal treatment of very short duration (from several nanoseconds to several milliseconds) that consists of the application of pulses with amplitudes from 100–300 V/cm to 20–80 kV/cm. This method induces the permeabilization of biological membranes by electrical piercing, which is called electroporation. The cell network maintains its capacity to act as a barrier for the passage of some undesired compounds, which improves extraction selectivity. Furthermore, plant materials treated with PEF seem to be less altered than thermally treated ones [54]. However, there are two main problems for its industrial application: the non-uniform nature of the ideal distribution of electric pulses and the limited variety of suitable solvents [73].

High voltage electrical discharge (HVED)-technology has also been recently studied for enhancing the extraction of bioactive compounds from different raw materials. HVED leads to the generation of hot, localized plasmas that emit high-intensity UV light, produce shock waves and bubble cavitation, and generate hydroxyl radicals from water photo-dissociation. In general, this technique provides a higher extraction rate than both PEF and ultrasound, but it may produce contaminants such as chemical products from electrolysis and free reactive radicals [88]. Some research has been published trying to scale up this technique to the pilot scale, but further research is still needed in order to achieve its industrial application [89].

These methods have been applied, at the lab scale, in the extraction of proteins from rapeseed press-cake [24], olive seeds [34], papaya seeds [54], and mango peels [26]. In some examples, they were employed as extraction techniques, while, in other cases, they were used as pretreatments followed by a solid–liquid extraction.

Proteins from mango peels were extracted by Parniakov et al. [26] using HVED and PEF, and the results were compared with those obtained by conventional aqueous extraction. HVED showed a higher protein yield than PEF and conventional aqueous extraction. Results obtained by conventional extraction improved when mango peels were pretreated with HVED and, especially, after PEF pretreatment. Very similar results were obtained when applying PEF and HVED for the extraction of proteins from papaya seeds [54]. Again, the protein yield obtained by conventional solid–liquid extraction significantly improved when the sample was pretreated with PEF.

A different approach was followed for the extraction of proteins from olive seeds [34]. In this case, HVED and PEF were used as pretreatments, followed by extraction with an aqueous:ethanol solution at different pHs. Extraction after HVED, using 23% ethanol at pH 12.0, provided the highest yield. Higher ethanol concentrations generated a smaller yield, probably due to the aggregation of proteins. In the case of the rapeseed press-cake [24], HVED was employed as the only step, and no subsequent solid–liquid extraction was performed. In this case, the use of powers higher than 240 kJ/kg did not improve the protein extraction yield. This fact was attributed to the release of oxygen reactive species that could react with proteins.

### 4.7. Extraction and Purification Using Nanomaterials

The use of nanomaterials in the extraction and purification of proteins is an interesting strategy to increase the sustainability of these steps. Nanomaterials present, at least, one of their dimensions in the nanoscale (1–100 nm). This is associated with extraordinary mechanical properties and enhanced electrical, magnetic, optical, thermal, or chemical properties. Moreover, they usually have good reactivity and can be easily functionalized. Different nanomaterials have been employed in the extraction and purification of proteins [90], although they have been scarcely applied in the case of fruit residues.

Dendrimers are a kind of nanomaterials with a structure similar to tree roots. They consist of layers called generations in which functional groups can be introduced. Carbosilane dendrimers are a special type of dendrimer that contain silicon atoms, have high stability and biocompatibility, and are easily functionalized. Carbosilane dendrimers can interact with proteins, and they have been applied for the purification and extraction of proteins from peach and plum seeds [74,75].

The use of carbosilane dendrimers functionalized with carboxylates groups under acid conditions has been found to result in the precipitation of proteins. This ability has enabled the development of a method for the purification of plum proteins as an alternative to acetone precipitation. The best yield was obtained with third generation dendrimers [74]. On the other hand, the ability of different cationic carbosilane dendrimers functionalized with amino, trimethylammonium, or dimethylamine groups to interact with proteins has also been studied. Second generation carbosilane dimethylamine-terminated dendrimers were selected for the purification of plum proteins. They resulted in the precipitation of 97% of proteins that could be precipitated with cold acetone [75]. The same authors, in other work, used single-wall carbon nanotubes coated with carbosilane dendrimers that were functionalized with sulphonate groups to extract proteins from plum seeds. Protein extraction yield was similar to the obtained with HIFU [76].

Moreover, gold nanoparticles functionalized with carbosilane dendrimers have been employed in the extraction of proteins from peach seeds [77]. Gold nanoparticles coated with dendrimers functionalized with sulphonate, carboxylate, or trimethylammonium groups were used and compared. The highest recovery of proteins was obtained with gold nanoparticles coated with second-generation carbosilane dendrimers functionalized with carboxylate groups at acid pH. Nevertheless, the recovery of proteins was low, and the strength of protein–dendrimer interactions was so high that very harsh conditions were required for their disruption.

## 5. Methods Used for the Release of Antioxidant Peptides

Once proteins have been extracted from a fruit residue, they must be submitted to a step to release peptides. There are three main approaches for this purpose: microbial fermentation with proteolytic microbes, proteolysis using enzymes from plants and microorganisms, and proteolysis using gastrointestinal enzymes. Table 2 shows the conditions employed to obtain peptides with antioxidant properties from fruit residues.

Commercial proteases can be expensive, and their industrial application might not be economically efficient. Microbial fermentation is a more environment-friendly and cost-effective proposal that has been tested in tomato seeds [36,37,38,39,40,41]. Different microorganisms were chosen: *Bacillus subtilis*, *Lactobacillus plantarum*, and a mixture of microorganisms from kefir culture. *B. subtilis* has been traditionally used to obtain fermented soybean products, while *L. plantarum* is a lactic acid bacteria that has been widely used in food production and preservation. In these cases, no previous extraction of proteins was carried out, and ground seeds were directly added to the culture. In another case, a kefir culture containing different lactic acid bacteria and yeasts was employed [37,41]. In all cases, fermentations took place at 37–40 °C and required very high times ranging from 20 to 24 h.

Alternatively, commercial proteases such as Alcalase, thermolysin, flavourzyme, protease P, Neutrase, trypsin, papain, and pepsin have been preferred in most works. Alcalase (the commercial name of subtilisin Carlsberg endopeptidase) is the most used enzyme for obtaining antioxidant peptides. This is not surprising, because Alcalase is a very cost effective food-grade protease with a low specificity that enables the release of a wide range of short peptides [43]. Trypsin, conversely, is a highly selective protease and has not been very effective in antioxidant peptide releasing [90]. Other food-grade enzymes such as pepsin, papain, pancreatin, and flavourzyme have also resulted in hydrolysates with high antioxidant activity [28,32,33,45,46,52,65,66]. Hydrolysis times were much lower than the required in microbial fermentation. Indeed, they usually ranged from, one-to-five hours, although there are some procedures that took longer. Moreover, hydrolysis must be carried out at a controlled pH and temperature conditions to obtain an optimum hydrolysis degree, which usually involves the careful optimization of these parameters.

Different strategies have been employed for the release of bioactive peptides from pumpkin oil cake proteins. Vaštag et al [58] compared the capacity of two different enzymes (Alcalase and flavourzyme) for this purpose. Alcalase showed a higher capacity to release peptides from pumpkin oil cake than flavourzyme (a hydrolysis degree (DH) of 53% for Alcalase and 37% for flavourzyme) and resulted in a hydrolysate with a higher antioxidant capacity. The DH increased up to 69% when the hydrolysate obtained using Alcalase was further hydrolyzed with flavourzyme, although antioxidant activity decreased [58]. Other enzymes employed for the release of peptides from pumpkin oil residues are pepsin, trypsin, protamex, and Neutrase [59,60,62,63]. However, many authors have pointed out that Alcalase is the most promising protease to produce pumpkin protein hydrolysates with an improved nutritional quality, but flavourzyme was the best to obtain antioxidant peptides. More recently, Nourmohammadi et al. [59] compared the feasibility of Alcalase to release antioxidant peptides with that of the trypsin enzyme. Again, Alcalase was preferred. Alternatively, other authors proposed the use of a protease working at an acid pH to obtain highly antioxidant peptides from pumpkin seeds [60].

Seeds from *Prunus* fruits (plum, peach, cherry, and apricot) also present a high protein content and have been exploited to obtain antioxidant peptides [11,14,27,28,29,33,35,55]. Like in other samples, different enzymes (Alcalase, flavourzyme, protease P, and thermolysin) were tried, and the highest antioxidant activity was observed in the hydrolysates obtained with Alcalase and thermolysin. Hydrolysis conditions were optimized for every enzyme, and although hydrolysis times from 2 to 4 h were the most common, some enzymes such as protease P required times up to 24 h. Moreover, the capacity of these seeds to release antioxidant peptides after simulated gastrointestinal digestion with pepsin and pancreatin was also studied [11], and the results demonstrated that peptides were less antioxidant than those obtained with previous enzymes. Moreover, the residue obtained after cherry wine fermentation, which was mainly constituted by cherry seeds, was simultaneously hydrolyzed with the Alcalase and Neutrase enzymes [55]. The resulting hydrolysate showed high antioxidant properties.

Seeds from other fruits such as olive, muskmelon, watermelon, tomato, bottle gourd, African breadfruit, and milled rapeseed were also used as sources of antioxidant peptides [16,17,21,23,32,43,47,48,50]. Different enzymes have been employed, but Alcalase and pepsin have generally reported the highest hydrolysis degree and antioxidant activity. In fact, the hydrolysis degree and antioxidant activity of hydrolysates obtained from olive seed proteins using the Alcalase enzyme was higher than the obtained with thermolysin, flavourzyme, Neutrase, and Trypsin [17]. Alcalase digestion also showed the highest hydrolysis degree and antioxidant activity in milled rapeseed proteins, as compared to pepsin, trypsin, subtilisin, and thermolysin [23]. Unlike them, watermelon seeds showed the highest hydrolysis degree when using the pepsin enzyme, while trypsin, Alcalase, papain, protease, pancreatin, and chymotrypsin yielded a lower hydrolytic activity [50,52]. Osukoya et al. compared the capacity to release peptides from the African breadfruit seed of pepsin enzyme with that of trypsin and pancreatin. In this case, pancreatin was the enzyme that resulted in the highest amount of peptides [56].

Jujube (also called red or Chinese date) and date palm also contain seeds with high protein contents that have been evaluated as sources of antioxidant peptides. Ambigaipalan et al. [65,66] employed different combinations of Alcalase, thermolysin, and flavourzyme enzymes for the hydrolysis of date palm seed proteins. They observed that the combination of Alcalase and flavourzyme resulted in the hydrolysate with the highest antioxidant activity [65]. In other research works, papain was employed for the hydrolysis of jujube seed proteins, and the results were compared with the obtained when using Alcalase and protease P; the hydrolysate obtained using papain yielded the highest antioxidant activity [45,46].

Hernández-Corroto et al. optimized the hydrolysis of pomegranate peel proteins using Alcalase and thermolysin, and they evaluated the potential of these proteins to release antioxidant peptides. No significant difference was observed in the antioxidant activity of the hydrolysates obtained by both enzymes [3,25]. Moreover, further studies revealed the contribution of phenolic compounds coextracted with proteins to the antioxidant activity that was observed in a hydrolysate obtained with thermolysin.

The performance of enzymatic hydrolysis can be improved by the previous treatment of proteins with ultrasound. Indeed, Wen et al. demonstrated that ultrasound treatment had a significant impact on proteins structure by improving their susceptibility to hydrolysis [51,53]. They submitted watermelon seed proteins to three different ultrasound treatments—single (20 kHz), dual (20 kHz/28 kHz), and tri-frequency (20/28/40 kHz))—before hydrolysis with Alcalase. All ultrasound treatments increased protein hydrophobicity by causing changes in secondary structure of proteins. Under most effective treatment (dual frequency ultrasound), the degree of hydrolysis and antioxidant activity increased [53].

## 6. Evaluation of Antioxidant Activity of Peptides 

Antioxidant activity, in a biological system, may occur through different mechanisms of action: (i) the inhibition of generation or the scavenging of ROS and reactive nitrogen species (RNS); (ii) the reduction of oxidants; (iii) the chelation of metals; (iv) as an antioxidant enzyme; and (v) the inhibition of oxidative enzymes [91]. Therefore, a comprehensive estimation of in vitro antioxidant activity must involve the use of different assays based on different mechanisms. Table 3 groups the assays employed for the evaluation of antioxidant activity in hydrolysates obtained from fruit residues.

Assays based on the scavenging of free radicals or oxidants have been the most commonly used to evaluate the antioxidant activity of hydrolysates. The two most common radical scavenging assays used non-biological oxidant species: ABTS (2,2’-azino-bis(3-ethylbenzothiazoline-6-sulfonic acid) and DPPH (1,1-diphenyl-2-picrylhydrazyl) [92] radicals. Assays using biological species such as nitric oxide and hydrogen peroxide have been less employed. All these assays consist of the spectrometric monitoring of their own radical or of a derived compound. Antioxidant capacity is estimated from the reduction in the absorbance of the oxidant or derived compound in the presence of potential antioxidants. A DPPH assay was employed to evaluate the antioxidant activity of peptides obtained by fermentation from tomato seeds. Fermentation with *B. subtilis* for 20 h resulted in an increase in DPPH activity from 23.5 to 68.5% [38]. Similarly, fermentation with kefir culture or *L. Plantarum* enabled a decrease from 41.24 and 40.89 to 10.84 and 4.95 µL/mg, respectively, in the IC₅₀ values obtained by the DPPH radical scavenging assay [36,37].

Other kind of methods have evaluated the capacity of potential antioxidants to inhibit an oxidation reaction. In some cases, these reactions result in the formation of free radicals such as superoxide radicals (O²⁻) or hydroxyl radicals (OH•) [3,11,14,16,17,25,27,28,33,35,47,55,65,66]; in others, by using different radicals (ROO•, HO•, O_2_•), a fluorescent probe is oxidized [51,65]. Superoxide radicals are formed by the oxidation of pyrogallol at basic pH. The antioxidant activity of the hydrolysates obtained from watermelon and Chinese cherry seeds was estimated by evaluating their capacity to inhibit this reaction and, thus, the formation of superoxide radicals [50,55]. On the other hand, hydroxyl radicals are generated by the Fenton reaction that is based on the oxidation of Fe^2+^ to Fe^3+^ in presence of H₂O₂. The formation of hydroxyl radicals is inhibited by the presence of antioxidant peptides from *Prunus* fruit, date, muskmelon, and olive seeds and from pomegranate peels [3,11,14,17,25,27,28,33,35,47,55,65,66]. A last assay in this category is based on the oxidation of fluorescein by the presence of ROS. The inhibition of this oxidation reaction is possible with peptides obtained from watermelon and date seeds [51,65].

Two methods have been employed to evaluate the reducing power of potential antioxidants in fruit residues: the ferric reducing antioxidant power assay (FRAP) and the ammonium phosphomolybdenum assay. Both are based on monitoring the capacity of peptides to reduce a probe cation (Fe^3+^ or Mo^6+^) [14,16,20,21,27,28,30,31,33,35,38,39,40,45,46,47,49,50,52,56,58,59,63,65]. The FRAP method, based on the reduction of Fe^3+^, has been the most employed by far.

Metal chelation activity has been explored in ferrous (FICA) and cupric (CICA) ion chelation activity assays. Both assays are based on the spectrometric measurement of colored complexes of Fe²⁺ and Cu²⁺ with ferrozine and pyrocatechol, respectively. The chelation of these metal ions by potential antioxidants avoids the formation of these colored complexes. These assays have been employed to evaluate the antioxidant activity of peptides released from watermelon, jujube, pumpkin, and date seeds [45,46,47,52,53,59,61,63,65].

The oxidation of biological molecules (lipids, proteins, and DNA) can result in the development of chronic diseases. For example, the oxidation of low density lipoproteins (LDL) cholesterol can result in atherosclerotic lesions, and the oxidation of DNA plays an essential role in the development of cancer. Moreover, the presence of ROS can result in the oxidative degradation of lipids through a process called lipid peroxidation, which can cause serious cell damage. Therefore, some antioxidant assays have been based on evaluating the capacity of peptides to inhibit oxidative damages on these molecules.

The lipid peroxidation inhibition of peptides and proteins from fruit residues has been measured using the ferric thiocyanate (FTC) assay and the thiobarbituric acid reactive substances (TBARS) assay. The FTC assay measures the primary products (hydroperoxides) formed during the oxidation of a fatty acid (e.g., linoleic acid or oleic acid). Hydroperoxides oxidize Fe^2+^ to Fe^3+^, and the latter forms a colored ferric thiocyanate complex. This method has been employed to evaluate the antioxidant activity in peptides obtained from *Prunus* fruits, African breadfruit, pumpkin, and olive seeds, as well as in proteins from watermelon seeds. The TBARS assay, on the other hand, measures a secondary product formed during lipid peroxidation (malondialdehyde) by its reaction with thiobarbituric acid. This assay was employed to evaluate the antioxidant activity in peptides released from pumpkin (*Cucubirta pepo* and *Cucurbita moschata*), watermelon, and wax gourd seeds [20,44,53,57]. The TBARS assay was also employed to evaluate the protection against oxidants of date seed protein hydrolysates using a biological model system—a cooked comminuted salmon. Hydrolysates obtained using Alcalase and flavourzyme and flavourzyme and thermolysin were able to achieve TBARS inhibition values of 32% and 30%, respectively, after seven days of storage compared with a positive control (butylated hydroxytoluene), which only achieved a 7% inhibition.

Two different assays, based on the inhibition of the oxidation of lipoproteins, have been employed to evaluate the antioxidant activity of peptides released from date seeds. One of them was based on the inhibition of β-carotene oxidation, and the other was based on the inhibition of low-density lipoprotein peroxidation. Date seed proteins were hydrolyzed using different combinations of Alcalase, thermolysin, and flavourzyme. The hydrolysate obtained with flavourzyme and thermolysin showed the highest inhibition of β-carotene oxidation, while the hydrolysate obtained with Alcalase and thermolysin yielded the highest inhibition of low-density lipoprotein peroxidation [66]. The same authors also employed an assay based on the protection of DNA molecules against oxidative damage. This assay evaluated the capacity of peptides to avoid the scission of supercoiled DNA strands in the presence of peroxyl and hydroxyl radicals. Peptides obtained by the hydrolysis of date seed proteins with Alcalase showed the highest capacity to inhibit DNA oxidative damage. Choudhary et al. [30] employed another assay that was also based on the inhibition of the oxidative degradation of DNA to study antioxidant activity of proteins from bottle gourd seeds. In this case, Cu²⁺ and H₂O₂ were employed to induce DNA oxidation.

All these methods are based on in vitro reactions, and though they are very useful as first screenings for potential antioxidants, further studies are required to confirm the real antioxidant capacity of peptides and proteins. Methods using cell cultures or those measuring antioxidant molecules in plasma or tissues from in vivo assays have been employed to confirm in vitro results.

Antioxidant activity has been evaluated by the determination of the level of intracellular ROS produced when cells are submitted to oxidative stress by the addition of a peroxide (hydrogen peroxide or tertbutylhydroperoxide). For that purpose, a fluorescence probe (2’, 7’-dichloro-dihydro-fluorescein diacetate) was employed. This molecule is hydrolyzed by intracellular esterase to form 2’, 7’-dichloro-dihydro-fluorescein. These molecules are next oxidized by intracellular ROS, which are produced under oxidative stress, and result in a highly fluorescent molecule (2’, 7’-dichloro-fluorescein) that can be monitored by fluorescence spectroscopy. This assay was employed to compare antioxidant capacity of peptides obtained from different genotypes of *Prunus* fruits and olive seeds. The measurement of ROS produced by cervical cancer cells (HeLa cells) under oxidizing conditions in the presence or absence of hydrolysates confirmed the antioxidant capacity of peptides, although no significant differences were observed among genotypes. This assay was also employed to measure antioxidant activity in five synthetic peptides found in watermelon seeds. The presence of a peptide with the Arg–Asp–Pro–Glu–Glu–Arg sequence reduced the generation of ROS in HepG2 cells under oxidizing conditions (H_2_O_2_).

Furthermore, ROS can increase cell membrane permeability, thus resulting in high intracellular concentrations of Ca²⁺. Intracellular Ca²⁺ concentration can be measured by fluorescence spectroscopy after adding a fluorescent dye. This assay demonstrated that the presence of peptide Arg–Asp–Pro–Glu–Glu–Arg reduced intracellular Ca²⁺ concentration in H₂O₂-damaged HepG2 cells and, thus, membrane cell damages generated under oxidizing conditions. Furthermore, an additional assay involving the use of two different DNA dyes has been employed to determine cell membrane oxidative damages. The acridine orange dye can penetrate into normal cell membranes and stain DNA into green, while ethidium bromide dye can penetrate into damaged cell membranes to stain DNA into orange or red. HepG2 cells under H₂O₂-induced-oxidizing stress showed less red cells when the Arg–Asp–Pro–Glu–Glu–Arg peptide was present [51].

Moreover, cancer cell initiation and progression has been linked to oxidative stress. In order to demonstrate that potential antioxidant peptides can decrease the proliferation of cancer cells and exert protective effects, cell viability can be determined using the MTT (3-(4,5-dimethylthiazol-2-yl)-2,5-diphenyltetrazolium bromide) assay. This assay measures the metabolic activity of cells through oxidation–reduction reactions happening in mitochondria via a succinate dehydrogenase system. The reduction of MTT in the mitochondria results in blue insoluble formazan that is measured by spectroscopy [16]. Studies on cell viability have demonstrated the lack of a cytotoxic effect of *Prunus* fruits and olive seed hydrolysates in normal HK-2 cells and their antiproliferative effect in malignant cells from human prostate cancer, colorectal adenocarcinoma, and cervical cancer. The MTT assay was also employed to demonstrate the cytoprotective effect of watermelon seed peptides with molecular weights below 1 kDa on RAW 264.7 cells submitted to oxidative stress induced by H₂O₂. The same authors also measured the nuclear factor erythroid 2-related factor 2 (Nrf2) and heme oxygenase-1 (HO-1) proteins, which participate in one of the most important antioxidant pathways in cells after treatment with H₂O₂. Nrf2 and HO-1 proteins were more expressed in cells when WSPHs-1 peptides from watermelon seeds were present [53].

The antioxidant activity of proteins has also been studied in some fruit residues. Different authors compared the antioxidant activity of fractions obtained by the application of the Osborne method to the extraction of proteins from bottle gourd [31], watermelon [49], pumpkin (*Cucurbita moschata*) [20], and melon [47] seeds. The fraction exhibiting the highest antioxidant activity in the case of the bottle gourd, watermelon, and pumpkin seeds was that containing globulins, while the glutelin fraction showed the lowest activity. As expected, proteins in globulin fractions were found to hold the highest amount of hydrophobic amino acids. The opposite behavior was observed for melon seed proteins. In this case, the globulin fraction yielded the least antioxidant activity, while the glutelin fraction showed the most and was also the one with the highest presence of polyphenols.

Previous studies have demonstrated that protein malnourishment lead to overoxidation [57]. Under these conditions, in vivo antioxidant system, consisting mainly of enzymes such as catalase, superoxide dismutase, glutathione peroxidase, and glutathione reductase, is not enough to avoid oxidative damage. The addition of antioxidants can increase the activity of these antioxidant enzymes. The antioxidant properties of proteins extracted from pumpkin seeds were investigated on low-protein fed rats submitted to CCl_4_ intoxication. The authors found that feeding of rats with pumpkin seed proteins resulted in increasing activity levels of catalase, superoxide dismutase, and glutathione peroxidase in the rat plasma and in a decreasing lipid peroxidation [57]. Nevertheless, pumpkin seed proteins could not inhibit the activity of the xanthine oxidase enzyme that promotes the generation of free radicals [61]. Moreover, different *Cucurbitaceae* (watermelon, bottle gourd, and pumpkin (*Cucurbita moschata*)) seeds yielded protein hydrolysates that were able to increase catalase levels in normal mice while also reducing in vivo lipid peroxidation [32]. Additionally, the synthetic peptide Arg–Asp–Pro–Glu–Glu–Arg, found in watermelon protein hydrolysates, was found to increase superoxide dismutase, glutathione peroxidase, and catalase activity in HepG2 cells under H₂O₂ induced oxidative damage [51].

## 7. Peptide Fractionation

In some cases, hydrolysates have been fractionated to obtain highly active fractions. The main easier fractionation technique is ultrafiltration. Ultrafiltration is a membrane separation technique that uses cut-off filters as molecular sieves (usually from 3 to 10 kDa); thus, peptides can be separated according to their molecular size. The main disadvantage of this technique is its low selectivity.

The use of ultrafiltration has been found to not provide a fraction with higher antioxidant peptides than the parent hydrolysate in the case of the olive, peach, and cherry seeds [17,28,33]. This is justified when considering that antioxidants work in a collaborative way, and, thus, the deficiency of a component in an antioxidant system can affect the efficiency of other. This behavior is common among antioxidant compounds. Nevertheless, in Chinese cherries and tomato seeds, ultrafiltration has provided fractions that exhibited the greatest antioxidant activity [21,55]. In most cases, most active peptides were in fractions under 3 and/or 1 kDa [17,21,35,53,55,93]. However, peptides above 5 kDa from peach and cherry seeds showed similar antioxidant activities to peptides below 3 kDa, with the peptides between 3 and 5 kDa being less effective [28,33]. The low selectivity of ultrafiltration filters may be related to this result [33].

Size-exclusion chromatography (SEC), dialysis, and preparative reversed-phase HPLC (RP-HPLC) are other techniques employed for the fractionation of hydrolysates. SEC is a low resolution technique normally followed by an orthogonal fractionation using semipreparative RP-HPLC. Peptides from Chinese cherry, tomato, and *Benincasa hispida* seeds have been fractionated using SEC with the dextran resins (Sephadex) and mobile phases consisting of a phosphate buffer (pH 6.5) [44], hydrochloric acid [39], or distilled water [55].

C18-bound phases have been always employed for the fractionation of peptides by RP-HPLC using elution gradients and mobile phases consisting of water with trifluoroacetic acid (TFA) (mobile phase A) and acetonitrile (ACN) with TFA (mobile phase B). TFA acts as ion-pairing reagent to maintain a low pH, create complexes with positively charged peptides, and minimize their ionic interactions with the hydrophobic stationary phase [94].

In many occasions, the combination of different fractionation techniques enables the isolation of most active peptides. Peptides from apricot seeds [27] were fractionated by dialysis followed by SEC in Sephadex G-25 and G-15, as well as RP-HPLC. Hydrolysates from Chinese cherry [55], watermelon [51], tomato [39], and wax gourd (*Benincasa hispida)* [44] protein seeds were fractionated by combining ultrafiltration, SEC, and RP-HPLC.

## 8. Peptide Identification

Antioxidant peptides obtained by extraction and proteolysis have been sequenced using mass spectrometry (MS). Usually, parent proteins have not been previously sequenced and they are not available in databases. In these cases, peptides are identified by de novo sequencing by the direct analysis of tandem mass spectra [3,14,16,17,25,33,35]. In a few cases, proteins from fruit wastes are present in databases, and a strategy based on database searching enables the identification of peptide sequence [21,29].

The identification of antioxidant peptides has enabled the observation of some common features within them, such as a high amount of hydrophobic (leucine/isoleucine, alanine, methionine, threonine, glycine, valine, and proline) [3,14,16,25,27,28,39,42,55] and aromatic amino acids (phenylalanine, tyrosine, tryptophan, and histidine) [3,14,16,21,25,27,28,39,40,47,55], as well as molecular weights below 1 kDa [14,16,25,27,28].

MS can directly provide information on the mass of a particular peptide but can also generate amino acid sequence information from tandem mass spectra (MS/MS). The MS analysis of peptides is possible using electrospray ionization (ESI) and matrix-assisted laser desorption/ionization (MALDI). Soft ionization allows for the transfer of polypeptide ions into the gas phase without their insource fragmentation. The identification of peptides by MS is carried out in the positive ion mode. In most cases, high resolution quadrupole time-of-flight (Q-TOF) or TOF mass analyzers are employed.

A MALDI source results mainly in singly charged ions and is considered a robust method of ionization in the presence of salts and detergents, much less prone to ionization suppression effects than ESI [94]. However, MALDI requires off-line sample deposition onto a target plate, and it is less convenient to couple with HPLC. Two peptides with molecular masses of 673.1 Da (Val–Leu–Tyr–Ile–Trp) and 566.9 Da (Ser–Val–Pro–Tyr–Glu) were identified in the most antioxidant fraction obtained from apricot seeds [27]. In other works with jujube and tomato seeds, most antioxidant peptides were observed in the mass range from 7 to 16 kDa in the case of jujube seeds [45] and from 0.5 to 0.8 Da and 1.2 to 1.5 Da in the case of tomato seeds [40], according to data obtained by MALDI-TOF. On the other hand, the molecular mass of “hispidalin” peptide from wax gourd seeds was 5.7 kDa, with the products of its hydrolysis being between 1.0 and 1.8 kDa [44].

Guo et al. [55] identified two peptides (Phe–Pro–Glu–Leu–Leu–Ile and Val–Phe–Ala–Ala–Leu) as main contributors to the antioxidant activity in Chinese cherry seed hydrolysate using just a Q-TOF MS and no coupling to a chromatographic separation. However, the analysis of hydrolysates by ESI-MS normally requires and is carried out by the previous separation of hydrolysates by HPLC. RP-HPLC is the most common chromatographic mode due to its high efficiency and the compatibility of mobile phases with ESI desorption. In this case, TFA, widely used as a counterion in the separation of peptides by RP-HPLC, is replaced by acetic acid or formic acid since TFA results in strong signal suppression in MS [3,16,17,33].

Hydrophilic interaction chromatography (HILIC) may also be useful for the separation of highly polar compounds that cannot be retained on RP-HPLC. The separation of analytes by HILIC is based on the interaction with a hydrophilic stationary phase like in normal-phase chromatography. However, HILIC uses water-miscible solvents (e.g., ACN), and elution is achieved by a water gradient that makes this technique suitable for coupling with MS. Both RP-HPLC and HILIC have been employed in the identification of peptides in plum, cherry, and peach seeds [28,33,35] and in pomegranate peel [25]. Figure 2 shows the total ion chromatogram obtained by RP-HPLC-ESI-QTOF and HILIC-ESI-Q-TOF of a hydrolysate obtained from plum seed proteins using Alcalase. Additionally, Figure 2 shows the mass spectrum of peptide His–Leu–Pro–Pro–Leu–Leu that was observed in both chromatographic modes. A comparison of peptides identified in plum, peach, and cherry seeds enabled the observation of some common sequences within *Prunus* fruits: Leu–Tyr–Ser–Pro–His, Leu–Tyr–Thr–Pro–His, Leu–Leu–Ala–Gln–Ala, Leu–Ala–Gly–Asn–Pro–Glu–Asn–Glu, Leu–Leu–Asn–Asp–Glu, and Leu–Leu–Met–Gln. The use of both chromatographic modes enabled, in all cases, a more comprehensive identification of peptides. Peptides identified using both chromatographic modes in hydrolysates obtained from pomegranate peel proteins extracted using HIFU, PLE, and DES were compared by Hernández-Corroto et al. [3,25]. Despite there being common peptides in the three hydrolysates, there were also many different peptides within them. This was attributed to the extraction of different kind of proteins by every technique [3,25].

Wen et al. employed a quadrupole-orbitrap MS instrument for the identification of most potent peptides in watermelon seeds. Five peptides were identified and then synthesized for their further characterization: Arg–Asp–Pro–Glu–Glu–Arg, Lys–Glu–Leu–Glu–Glu–Lys, Asp–Ala–Ala–Gly–Arg–Leu–Gln–Glu, Leu–Asp–Asp–Asp–Gly–Arg–Leu, and Gly–Phe–Ala–Gly–Asp–Asp–Ala–Pro–Arg–Ala [51]. The peptide Arg–Asp–Pro–Glu–Glu–Arg showed the highest antioxidant activity, according to results obtained by ABTS, DPPH, and oxygen radical antioxidant capacity (ORAC) assays. Moreover, it also showed cytoprotective effects on HepG2 cells under induced oxidative stress.

The introduction of nanoelectrospray (nanoESI) has enabled an increase in sensitivity. Peptides in most antioxidant fractions from fermented tomato seeds were identified by nLC/MS/MS (QTOF), obtaining many sequences below 600 Da and 5–6 amino acid residues. Among peptides, GQVPP showed a significant antioxidant activity (97% DPPH scavenging activity at 0.4 mM) [39].

None of the bioactive peptides cited in this section of the review were previously recorded in the BIOPEP database.

## 9. Conclusions

Antioxidant peptides have been discovered in some fruit residues (largely seeds and peels). Most of them have been isolated by solid–liquid conventional extraction under basic conditions. Ultrasound-assisted extraction, pressurized liquid extraction, microwave-assisted extraction, and the application of electrical energy (pulsed electric fields or high-voltage electrical discharges) have been employed in order to improve the sustainability and yield of extraction. An increasing trend is the use of non-polluting solvents such as deep eutectic ones. Future progress in this area will likely be focused in the combination of different (orthogonal) extraction techniques and strategies. Some works have already demonstrated that when using a combination of DES with HIFU or MAE, orpretreatment with HVED or PEF. the extraction of protein accomplishes higher yields—however, further research in this direction is required to attain solid developments in this field.

While non-conventional methods are promising because of their many advantages (such as the reduction of processing time, the use of greener and/or safer solvents, and higher extraction yields), most of them are not ready for scaling up in the industry. Extraction and purification with carbosilane dendrimers is another worthwhile proposal, although it has not been studied much yet.

Some antioxidant peptides were directly extracted from these residues, but most were found in a latent state as part of protein sequences. The release of these peptides was mainly achieved by enzymatic digestion, with Alcalase being the enzyme that usually resulted in higher antioxidant peptides. The use of unrelated in vitro assays based on different antioxidant mechanisms guarantees a comprehensive evaluation of a fruit residue’s peptide capacity. The evaluation of the capacity of peptides to scavenge free radicals, inhibit lipid peroxidation, or reduce oxidants is the most popular study area, although further experiments using cell cultures and animal models are essential to confirm in vitro results. The fractionation of peptides by ultrafiltration, size-exclusion chromatography, and reversed-phase chromatography has proven useful to isolate most antioxidant peptides. Tandem mass spectrometry has enabled the identification of peptide sequences with significant antioxidant properties that, in some cases, have been synthesized for further study. In general, antioxidant peptides from fruit seeds have shown short sequences and contain hydrophobic and aromatic amino acids.

The research discussed in this review shows that fruit residues store great amounts of antioxidant peptides, which are highly valuable products with promising applications as nutraceuticals or added to functional foods. As such, harvesting these biomolecules may represent a partial solution to the increasing environmental concerns about the management of fruit residues.

## Figures and Tables

**Figure 1 foods-09-01018-f001:**
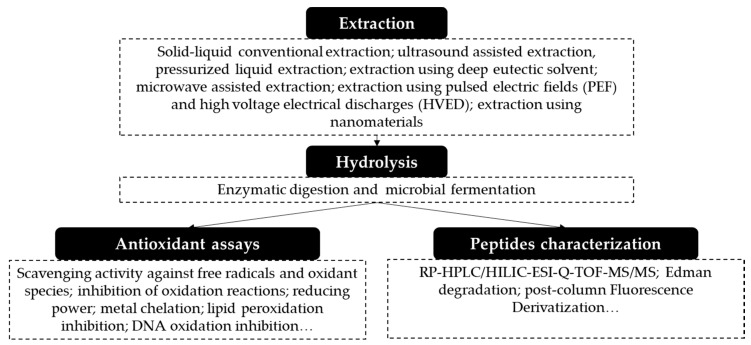
General procedure employed for obtaining proteins and peptides from fruit residues. Rp-HPLC: reversed-phase HPLC; HILIC: hydrophilic interaction chromatography; ESI: electrospray ionization; and Q-TOF: quadrupole time-of-flight.

**Figure 2 foods-09-01018-f002:**
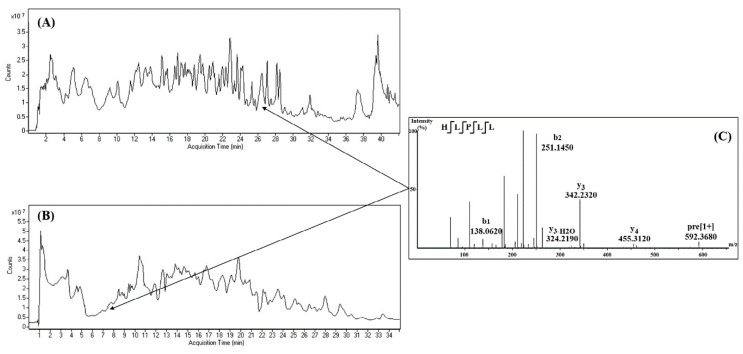
Total ion chromatograms corresponding to a hydrolysate obtained from plum seed proteins using Alcalase hydrolysate obtained by RP-HPLC (**A**), HILIC (**B**), and MS/MS spectrum of a peptide simultaneously observed in both chromatographic modes (**C**). With permission from [35].

**Table 1 foods-09-01018-t001:** Methods employed in the extraction and purification of proteins from fruit residues.

Residue	Extracting Media	Extraction Conditions	Protein Purification Conditions	Refs.
SOLID–LIQUID CONVENTIONAL EXTRACTION				
Tomato seeds	NaCl (1.5%, pH 11.5)	DGS: extracting medium at ratio 1:10 (w/v); stirring (room temperature and 1 h)	Centrifugation and precipitation at pH 4.0	[21,43]
Tomato seeds	NaOH aq. (pH 7.5–1.5)	1 g of DGS and 82 mL water; stirring (50 °C and 50 h)	Centrifugation and filtration	[37,41]
Tomato seeds	NaOH aq. (pH 7.5)	DGS: extracting medium at ratio 1:82.81 (30 °C and 50 h)	Centrifugation and precipitation at pH 3.9	[42]
Watermelon seeds	Alkali (0.8%)	DGS: extracting medium at ratio 1:30 (40 °C and 30 min)	Precipitation at pH 4.5	[52]
Watermelon seeds	NaOH aq. (pH 12.0)	DGS: extracting medium at ratio 1:10 (w/v); stirring (1 h); 2 additional extractions after centrifugation	Precipitation at pH 4.0	[48,50]
Jujube seeds	(I) Water; (II) 50 mM Tris-HCl (pH 7.5); (III) 0.6 M NaCl (0.1% HCl); and (IV) acetic acid (5%)	(I and II) 1.5 g ground seeds and 20 mL solvent; stirring (4 °C and 2 h); (III) Idem at ground seeds:solvent 1:3 (w/v); and (IV) 5 g and 20 mL solvent. Shaking overnight (80 rpm)	(I, II, and III) Precipitation with (NH₄)₂SO₄ and dialysis (24 h, 4 °C); (IV) filtration and precipitation with acetone	[45]
Chinese cherry seeds	NaOH aq. (pH 10.0)	DGS: extracting medium at 1:20 (w/v); stirring (40 °C and 40 min)	Filtration and precipitation at pH 3.84	[55]
African breadfruit seeds	NaOH aq. (pH 9.0)	DGS: extracting medium at ratio 1:10 (w/v); stirring (30 min at room temperature)	Precipitation at pH 4.5	[56]
Pumpkin (*Cucurbita pepo*) seeds	NaOH aq. (pH 10.0)	DGS	Precipitation at pH 5.0	[57,61]
Pumpkin (*Cucurbita pepo*) oil cake	NaOH aq. (pH 10.0)	DGS	Filtration and precipitation at pH 5.0	[58,62]
Pumpkin (*Cucurbita pepo*) oil cake	NaOH aq. (pH 10.0)	Defatted oil cake: extracting media at ratio 1:10 (w/v)	Precipitation at pH 5.0	[59]
Pumpkin (*Cucurbita pepo*) oil cake	NaOH aq. (pH 11.0)	DGS: extracting medium at 1:30 (w/v); stirring (50 °C and 1.5 h)	Precipitation at pH 5.3	[60]
Pumpkin (*Cucurbita moschata*), watermelon, and bottle gourd seeds	Tris HCl (pH 8.0)	200 mg of DGS and 50 mL buffer (1 h)	Centrifugation and precipitation with acetone	[32]
Bottle gourd seeds	50 mM phosphate buffer (10 mM EDTA, 100 mM KCl, 1 mM DTT, and 1% SDS)	Sample: extracting medium at ratio 1:3 (w/v) (3 times)	Filtration and precipitation with chilled ethanol	[30]
Wax gourd seed	20 mM phosphate buffer (pH 6.5, 5.0 mM EDTA, and 10 mM DTT) (buffer I) and phosphate (2.0 mM EDTA, 1 mM DTT, urea 4 M, and 2% Triton X 100) (buffer II)	500 g of DGS and buffer I, 3 h; centrifugation and 2nd extraction under same conditions; centrifugation and 3rd extraction with buffer II and centrifugation	Dialysis, centrifugation, and filtration	[44]
Jujube seeds	Tris-HCl (pH 7.5)	1.5 g of ground seeds and 50 mM buffer (2 h)	Centrifugation, precipitation with (NH₄)₂SO₄ (4 °C), centrifugation, and dialysis	[46]
Milled rapeseed	50 mM Tris-HCl (pH 8.5, 750 mM NaCl, 5 mM EDTA, and 0.3% Na₂O₅S₂)	0.1 g/L (room temperature, 1 h)	Centrifugation	[23]
Pumpkin (*Cucurbita moschata*) seeds	Osborne method: (1) water; (2) Tris-HCl (5% NaCl); (3) isopropanol (55%); and (4) acetic acid (0.2 N)	150 mg of DGS and 10 mL extracting medium (60 min). Next extractions at extracting medium:sample ratio 7:1	(I, II, and III) Centrifugation and precipitation with acetone	[20]
Muskmelon seeds	Osborne method: (1) water; (2) NaCl (5%); (3) NaOH (0.1 M); and (4) ethanol (70%)	100 g of DGS and 500 mL extracting medium (60 min)	(I) Centrifugation; (II) centrifugation and dialysis; (III) precipitation at pH 4.0; and (IV) evaporation at 40 °C	[47]
Bottle gourd seeds	Osborne method: (1) water; (2) Tris-HCl (100 mM, pH 8.1, 0.5 M NaCl); (3) isopropanol (55%); and (4) acetic acid (0.2 N)	8 g of DGS and 60 mL extracting medium (60 min). Next extractions at extracting medium:sample ratio 7:1	(I, II, III, and IV) Centrifugation and precipitation with acetone	[31]
Watermelon seeds	Osborne method	-	-	[49]
ULTRASOUND-ASSISTED EXTRACTION				
Pomegranate peel	100 mM Tris-HCl (pH 7.5, 0.5%, SDS, and 0.25% DTT)	150 mg milled peel and 5 mL buffer using HIFU (30%, 1 min)	Evaporation and precipitation with cold acetone	[3]
Peach, plum, apricot, cherry, and olive seeds	100 mM Tris-HCl (pH 7.5, 0.5% SDS, and 0.5% DTT)	30 mg DGS and 5 mL buffer using HIFU (30%, 1 min)	Precipitation with cold acetone	[11,13]
Plum seeds	100 mM Tris-HCl (pH 7.5, 1% SDS, and 0.25% DTT)	30 mg DGS and 5 mL buffer using HIFU (30%, 1 min)	Precipitation with cold acetone and filtration	[14,35]
Olive and peach seeds	100 mM Tris-HCl (pH 7.5, 0.5% SDS, and 0.5% DTT)	30 mg DGS and 5 mL buffer using HIFU (30%, 5 min)	Precipitation with cold acetone	[16]
Olive seeds	125 mM Tris-HCl (pH 7.5, 1% SDS, and 0.1% DTT)	30 mg milled seeds and 5 mL buffer using HIFU (30%, 5 min)	Precipitation with cold acetone	[17]
Plum and peach seeds	50 mM Tris-HCl (pH 7.4) and 15 mM NaCl (buffer I)50 mM Tris-HCl (pH 7.4) and 15 mM NaCl and 1% SDS and 25 mM DTT (buffer II)	200 mg DGS and 10 mL of buffer I or II (10 min) and shaking (overnight)	Evaporation and precipitation with cold acetone	[29]
Cherry seeds	100 mM Tris-HCl (pH 7.5, 1% SDS, and 0.5% DTT)	30 mg DGS and 5 mL buffer using HIFU (30%, 5 min)	Precipitation with cold acetone	[33]
PRESSURIZED LIQUID EXTRACTION				
Pomegranate peel	Ethanol (70% (v/v))	2 g ground dried peels and 8 g sand (1500 psi; 120 °C; static extraction time, 3 min; extraction time, 12 min; static cycles, 1)	Evaporation and precipitation with cold ethanol	[25]
EXTRACTION USING DEEP EUTECTIC SOLVENTS				
Pomegranate peel	Choline chloride:AA:H₂O in 1:1:10 molar ratio	150 mg dried peels and 5 mL extracting medium (HIFU, 11 min, and 30%)	Evaporation and precipitation with cold ethanol	[25]
ULTRASOUND–MICROWAVE SYNERGISTIC EXTRACTION				
Pumpkin (*Cucurbita moschata*) seeds	PEG 200-choline chloride at 3:1 molar ratio	Microwave-assisted extraction (6 min, 120 W); ultrasound-assisted extraction (30 min, 240 W); water bath extraction (43 °C and 60 min); ultrasound–microwave synergistic extraction (28%, 28 g /L, 140 W, 43 °C and 4 min)	Isoelectric point precipitation; ethanol precipitation; centrifugation; centrifugation, isoelectric point precipitation, and ethanol precipitation	[64]
EXTRACTION USING PULSED ELECTRIC FIELD (PEF) AND HIGH VOLTAGE ELECTRICAL DISCHARGE (HVED)				
Olive seeds	Water	Sample: extracting medium at ratio 10 (w/w); pretreatment with HVED, PEF, and ultrasound; extraction (2 min and 150 rpm)	-	[34]
Rapeseed press-cake	Water	Sample: extracting medium at ratio 20 (v/w); HVED (240 kJ/kg)	-	[24]
Mango peels	I and II) Water; III) water (pH 11.0); IV) water and water (pH 11.0)	300 g of sample at ratio 1/10 (w/v); (I) PEF (13.3 kV/cm, 0.5 Hz); (II) HVED (40 kV/cm, 0.5 Hz); (III) aqueous extraction (20–60 °C and pH = 2.5, 6.0, 11.0); (IV) PEF; and (I) and aqueous extraction (50 °C, pH 6.0, and 3 h)	-	[26]
Papaya peels	(I and II) Water; (III) water (pH 11.0); (IV) water and water (pH 11.0)	300 g sample at ratio 1:10 (w/v); (I) PEF (13.3 kV/cm, 0,5 Hz); (II) HVED (40 kV/cm, 0.5 Hz); (III) aqueous extraction (20–60 °C, pH = 2.5, 6.0, 11.0); (IV) PEF; and (I) aqueous extraction (50 °C, pH 7.0, and 3 h)	-	[54]
METHODS USING NANOMATERIALS				
Plum seeds	100 mM Tris–HCl buffer (pH 7.5, 1% SDS, and 0.25% DTT)	30 mg DGS and 5 mL buffer using HIFU (30%, 1 min)	3G carboxylate-terminated dendrimers at pH 1.8 (30 min)	[74]
Plum seeds	100 mM Tris–HCl buffer (pH 7.5, 1% SDS, and 0.25% DTT)	30 mg DGS and 5 mL buffer using HIFU (30%, 1 min)	2G dimethylamino-terminated dendrimers at pH 7.5 (30 min)	[75]
Plum seeds	-	3G single wall carbon nanotubes functionalized with sulphonate-terminated carbosilane dendrimers at pH 7.5 with shaking (1 h)	Ultrafiltration	[76]
Peach seeds	-	2G gold nanoparticles coated with carbosilane dendrimers with carboxylate groups at pH 2.5 with shaking (2 h)	Ultrafiltration	[77]

Note: DGS: defatted ground seeds; Tris: tris(hydroxymethyl)aminomethane; DTT: dithiothreitol; SDS: sodium dodecyl sulfate; HIFU: high-intensity focused ultrasound.

**Table 2 foods-09-01018-t002:** Conditions employed to obtain antioxidant peptides from fruit residues.

Fruit Residue	Enzyme/Microorganisms	Buffer (pH)	Temperature (°C)	Time (h)	Refs.
RELEASE OF PEPTIDES BY MICROBIAL FERMENTATION					
Tomato seed	*Lactobacillus plantarum*	—	37	24	[36]
Tomato seed	Water kefir microbial mixture	—	37	24	[37,41]
Tomato seed	*Bacillus subtilis*	—	40	20	[38]
Tomato seed	*Bacillus subtilis*	—	37	24	[39,40]
RELEASE OF PEPTIDES BY ENZYMATIC DIGESTION					
Pumpkin oil cake	Alcalase	Tris-HCl (0.1 M and pH 8.0)	50	1	[58]
	Flavourzyme			1	
	Alcalase and flavourzyme			2	
Pumpkin oil cake	Alcalase	Phosphate (pH 8.0)	50	0–2.5	[62]
	Flavourzyme	Phosphate (pH 7.0)	50		
	Pepsin	Phosphate (pH 3.0)	37		
Pumpkin oil cake	Alcalase	Tris-HCl (pH 9.0)	50	3.5	[59]
	Trypsin	Tris-HCl (pH 8.0)	35	5	
Pumpkin seed	Acid protease	pH 2.5	50	5	[60]
Pumpkin meal	Alcalase	pH 8.0	55	5	[63]
	Flavourzyme	pH 7.0	50		
	Protamex	pH 6.5	50		
	Neutrase	pH 7.0	50		
Peach, plum, apricot, and olive seeds	Pepsin and pancreatin	pH 2.0 and pH 8.0	37	3	[11]
Apricot seeds	Alcalase	Borate (5 mM and pH 8.5)	50	4	[11]
	Thermolysin	Phosphate (5 mM and pH 8.0)		4	
	Flavourzyme	Phosphate (5 mM and pH 7.5)		8	
Plum seeds	Alcalase	Borate (5 mM and pH 8.5)	50	3	[14,35]
	Thermolysis	Phosphate (5 mM and pH 8.0)	50	4	
	Flavourzyme	Phosphate (5 mM and pH 7.0)	50	7	
	Protease P	Phosphate (5 mM and pH 7.5)	40	24	
Apricot seeds	Alkaline and flavor proteases	-	-	-	[27]
Peach seeds	Alcalase	Phosphate (5 mM and pH 8.0)	50	4	[28]
	Thermolysin	Phosphate (5 mM and pH 8.0)	50	4	
	Flavourzyme	Ammonium bicarbonate (5 mM and pH 6.5)	50	3	
	Protease P	Phosphate (5 mM and pH 7.5)	40	7	
Cherry seeds	Alcalase	Borate (pH 8.5)	50	7	[33]
	Thermolysin	Phosphate (pH 8.0)			
	Flavourzyme	Bicarbonate (pH 6.0)			
Chinese cherry seeds	Alcalase and Neutrase	Water (pH 7.5)	50	2	[55]
Olive and peaches seeds	Alcalase	Borate (5 mM and pH 8.5)	50	4	[16]
Olive seeds	Alcalase	Phosphate (5 mM and pH 8.0)	50	2	[17]
	Thermolysin	Phosphate (5 mM and pH 8.0)			
	Flavourzyme	Ammonium bicarbonate (5 mM and pH 6.0)			
	Trypsin	Tris-HCl (5 mM and pH 9.0)			
	Neutrase	Phosphate (5 mM and pH 7.0)			
Tomato seeds	Alcalase	Phosphate (pH 8.0)	50	0.5–3	[43]
Tomato seeds	Alcalase	Phosphate (pH 8.0)	50	2.3	[21]
Milled rapeseed	Pepsin	Phosphate (0.1 M and pH 2.0)	40	3	[23]
	Trypsin	Phosphate (0.1 M and pH 7.0)	40	3	
	Alcalase	Phosphate (0.1 M and pH 7.0)	50	3	
	Subtilisin	Phosphate (0.1 M and pH 8.0)	60	3	
	Thermolysin	Phosphate (0.1 M and pH 8.0)	60	24	
Jujube seeds	Papain	Tris-HCl (50 mM and pH 6.5–7.5)	65	1.5	[45,46]
	Alcalase	Tris-HCl (50 mM and pH 6.5–8.5)	60		
	Protease P	Tris-HCl (50 mM and pH 7.5)	37		
Muskmelon seeds	Pepsin and Trypsin	pH 2.0 and pH 7.0	37	6	[47]
Pumpkin (*Cucurbita moschata*), watermelon, and bottle gourd seeds	Trypsin	Tris-HCl (50 mM and pH 7.5)	-	4	[32]
Watermelon seeds	Alcalase	Phosphate (5 mM and pH 8.0)	60	5	[48,50]
	Trypsin	Phosphate (5 mM and pH 8.0)	37		
	Pepsin	Glycine (5 mM and pH 2.2)	37		
Watermelon seeds	Alcalase	pH 8.5	55	3	[51]
Watermelon seeds	Papain	—	—	—	[52]
	Pepsin	pH 2.4	37	3	
	Protease	—	—	—	
	Pancreatin	—	—	—	
	Trypsin	—	—	—	
	Chymotrypsin	—	—	—	
Watermelon seeds	Alcalase	NaOH aq. (pH 9.0)	50	0.8	[53]
African breadfruit seeds	Trypsin, pepsin, and pancreatin	Water	-		[56]
Date palm seeds	Alcalase	pH 8.0	50	1	[65,66]
	Flavourzyme	pH 7.0		2	
	Thermolysin	pH 8.0		3	
Pomegranate peel	Alcalase	Borate (5 mM and pH 9.0)	50	2	[3]
	Thermolysin	Phosphate (5 mM and pH 7.5)	70	1	
Pomegranate peel	Alcalase Thermolysin	Borate (5–10 mM and pH 9.0) Phosphate (5 mM and pH 7.5) or borate (100 mM and pH 7.5)	50 70	2 1	[25]

Note: Enzymes in bold characters are the ones which yielded the higher quantity of antioxidant peptides in each research work.

**Table 3 foods-09-01018-t003:** Assays employed for the evaluation of antioxidant activity.

Assay	Methodology	Refs.
EVALUATION OF THE CAPACITY TO SCAVENGE FREE RADICALS AND OXIDANT SPECIES		
Scavenging effect on hydrogen peroxide (H₂O₂) radicals	Measurement of the reduction in the absorbance of a H₂O₂ solution at 230 nm after incubation with potential antioxidants.	[20,31,48]
Scavenging effect on ABTS (2,2’-azino-bis(3-ethylbenzothiazoline-6-sulfonic acid) radicals	ABTS radicals that absorb at 734 nm are produced by the reaction of ABTS with potassium persulfate. The method evaluates the reduction in the absorbance of ABTS radicals due to the presence of potential antioxidants.	[3,11,14,16,17,20,24,25,28,31,33,34,35,36,37,41,42,45,46,49,51,52,54,55,58,62,65]
Scavenging effect on nitric oxide (NO) radicals	Nitric oxide radicals are formed from nitroprusside and the incubation of formed nitric oxide radicals with a Griess reagent (1% sulphanilamide, 2% H_3_PO_4_, and 0.1% naphthylethylene diamine dihydrochloride) results in nitrite ions. Nitrite ions can be measured by the formation of a compound that absorbs at 546 nm. The scavenging of nitric oxide radicals by potential antioxidants reduces nitrite ion formation and their absorbance at 546 nm.	[20,31,49]
Scavenging effect on DPPH (1,1-diphenyl-2-picrylhydrazyl) radicals	Measurement of the decrease in the absorption of DPPH radicals at 515–517 nm when potential antioxidants are added.	[14,17,20,21,26,27,30,31,34,36,37,38,39,40,41,42,43,44,45,46,47,48,49,51,52,53,54,55,56,59,60,61,63,65]
EVALUATION OF THE CAPACITY TO INHIBIT OXIDATION REACTIONS		
Inhibition of formation of superoxide (O²⁻) radicals	The assay measures the rate of pyrogallol autooxidation in presence and absence of potential antioxidants at 320–420 nm.	[50,55]
Inhibition of formation of hydroxyl (OH•) radicals	Hydroxyl radicals are generated by the oxidation of Fe^2+^ to Fe^3+^ in the presence of H₂O₂. The presence of Fe^2+^ is monitored by the formation of a complex with 1,10-phenanthroline that absorbs at 536 nm. The presence of potential antioxidants inhibits the oxidation of Fe^2+^ and results in an absorbance increase.	[3,11,14,16,17,25,27,28,33,35,47,55,65]
Oxygen radical antioxidant capacity (ORAC)	The method is based on the oxidation of fluorescein by reactive oxygen species (ROS) resulting from the radical initiator 2,2’-azobis(2-methylpropionamidine) dihydrochloride. The inhibition of fluorescein oxidation by the presence of potential antioxidants is measured from the increase in fluorescence intensity.	[51,65]
EVALUATION OF THE REDUCING POWER		
Ferric reducing antioxidant power (FRAP)	Measures the ability of potential antioxidants to reduce Fe³⁺ from the ferricyanide complex to Fe²⁺-complex. Formation of Fe²⁺-complex is measured at 700 nm.	[14,16,20,27,28,30,31,33,35,38,39,40,45,46,47,49,50,52,56,58,62,63,65]
Ammonium phosphomolybdenum	The method evaluates the capacity of potential antioxidants to reduce Mo⁶⁺ to Mo⁵⁺. Presence of Mo⁵⁺ is monitored by the subsequent formation of a green phosphor/Mo⁵⁺ complex that absorbs at 695–65 nm.	[20,21,30,31,59]
EVALUATION OF THE METAL QUELATION ACTIVITY		
Ferrous ion chelation activity (FICA)	Ferrozine reacts with Fe²⁺ to form a complex that absorbs at 562 nm. In the presence of chelating agents, the complex is disrupted, resulting in a decrease in absorption at 562 nm.	[45,46,52,53,59,61,63,65]
Cuprous ion chelation activity (CICA)	Reaction of pyrocatechol and Cu²⁺ results in a substance that absorbs at 632 nm. The presence of a metal chelator disrupts this molecule and reduces the absorbance.	[47]
EVALUATION OF THE CAPACITY TO INHIBIT LIPIDS AND LIPOPROTEINS OXIDATION		
Ferric thiocyanate	Primary products resulting from the oxidation of linoleic acid are incubated with EtOH, ammonium thiocyanate, and FeCl₂, leading to the formation of Fe(SCN)²⁺ that absorbs at 500 nm. Presence of potential antioxidants results in the inhibition of linoleic acid oxidation and the reduction of absorption.	[14,16,17,28,33,35,44,49,56,63]
Thiobarbituric acid reactive substances (TBARS)	The presence of secondary oxidation products formed during oxidation of linoleic acid is measured by the reaction of one of them, the malondialdehyde, with SDS, acetic acid, and TBA at 532 nm. The presence of potential antioxidants reduces this absorbance.	[20,23,31,32,44,53,57,66]
β-carotene linoleate	It measures the ability of potential antioxidants to decrease the oxidative bleaching of *β*-carotene in an oil-in-water emulsion. The reaction is monitored by measuring the absorbance at 470 nm immediately after the addition of a potential antioxidant.	[66]
Inhibition of Cu²⁺-induced low-density lipoprotein (LDL) peroxidation	This assay measures the peroxidation induced by cupric sulfate in LDL. Presence of potential antioxidants results in the inhibition of the oxidation and the reduction of the absorbance of conjugated dienes at 344 nm.	[66]
EVALUATION OF THE CAPACITY TO INHIBIT DNA OXIDATION		
Supercoiled-to-Nicked-Circular-Conversion (SNCC)	Oxidation of supercoiled DNA into nicked circular DNA in the presence of Cu²⁺ and H₂O₂ is monitored by measuring the fluorescent intensity of ethidium-stained nicked circular DNA. The presence of a potential antioxidant inhibits this reaction, and the signal corresponding to the oxidized form of DNA decreases.	[30]
Inhibition of peroxyl and hydroxyl radical-induced supercoiled strands scission	Strand scission of supercoiled DNA is measured in the presence of peroxyl and hydroxyl radicals. After incubation, DNA is separated by gel electrophoresis, and the intensity of supercoiled DNA bands in the presence and absence of potential antioxidant are compared.	[66]
EVALUATION OF THE CAPACITY TO INHIBIT OXIDATIVE DAMAGE INDUCED IN CELLS		
2’, 7’-dichloro-dihydro-fluorescein diacetate (DCFH-DA) fluorescent probe	Oxidative stress in cells is induced by the addition of a strong oxidant (H₂O₂ or other peroxide). DCFH-DA fluorescence probe, added to cell culture, reacts with ROS to produce fluorescent DCF that is measured at an λexcitation of 488 nm and an λemission of 585 and 530 nm. The presence of a potential antioxidant inhibits ROS generation and DCF signal decreases.	[16,51]
Intracellular concentration of Ca²⁺ determination	Intracellular Ca²⁺ is measured with fluorescent dye Fura-2 AM. Fura-2AM is cleaved by intracellular esterase, and the resulting Fura-2 can bind to Ca²⁺ and cause strong fluorescence under a 330–350 nm excitation light. Fluorescence intensity decreases in H₂O₂-damaged cells treated with potential antioxidants.	[51]
Acridine orange/ethidium bromide (AO/EB) fluorescent staining	Cell membrane damage is measured by evaluating the staining of DNA with EB or AO using an inverted fluorescence microscope. The presence of potential antioxidants will reduce the number of red cells resulting from the staining with EB and will increase the number of green cells resulting from the staining with AO.	[51]

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
