# Peer review of "Extraction and Characterization of Antioxidant Peptides from Fruit Residues"

_foods, 2020, doi:10.3390/foods9081018_

Round 1
Reviewer 1 Report
In recent years, there has been a considerable interest in bioactive peptides derived fromfood proteins which might have beneficial effects on human health. In this publication, the authors have drawn attention to the potential of using fruits residues to obtain bioactive peptides. In addition, the extraction of bioactive substances from valuable by-products of the food industry is in the zero waste trend. The publication discusses briefly antioxidant peptides characteristics from fruits residues as well as methods of their extraction and purification. The authors have extensively described the most frequently used extraction methods, citing many examples for different raw materials. Also discussed were some of the innovative technologies such as: extraction using pulsed electric fields (PEF) and high voltage electrical discharges (HVED). The data in the publication are presented in 2 extensive tables, which increased the clarity publication.
Author Response
Thank you very much for you review report. We are glad of your positive report.
Reviewer 2 Report
Foods-842562- Extraction and characterization of antioxidant peptides from fruit residues
This review describes the most recent research findings in relation to the methods used for extraction of proteins from fruit residues and the generation, fractionation and characterisation of antioxidant peptide derived therefrom. Overall, the review is quite comprehensive and may be useful to food scientists and manufacturers. However, in general, the manuscript appears to be a compilation of literature, not a true "review" from the perspective of providing any critical review of the literature, useful comments or insights into the topic. The authors need to provide comments and conclusions which they draw from their accumulation of evidence.
Other comments are given in below:
The authors lists the chemicals/reagents used in the extraction of proteins from fruit residues and indicate that these are ‘polluting reagents and volatile organic solvents’. However, among other applications the intended use for the fruit residue-derived proteins and peptides are as functional food ingredients. The authors need to indicate that some of the reagents are not food-grade (e.g., Urea, SDS, DTT, mercaptoethanol) and if the intended applications are for human/animal consumption then alternative extractants are required. Furthermore, they need to comment on how applicable the methodology used is to a food industry environment.
The authors indicate that ‘Protein extraction requires the breakdown of tissues, cell membranes, and cell walls in order to release the intracellular material. In the case of plant tissues, the difficulty is high due to the presence of large vacuoles, the rigidity and thickness of cell walls, and the heterogeneity of proteins.’ However, there is limited information on the state of the raw material prior to protein extraction, is it wet/dry, is it milled etc, what is the best format for extraction and is this feasible on an industry basis.
A paragraph describing the quantities, types and characteristics of proteins in the fruit residues (and potentially a SDS PAGE profile of the proteins) should be inserted in section 4 ‘Techniques used in the extraction and purification of proteins’ following paragraph 1 to provide the reader with basic background information before the author delves into the methods used for their extraction.
The author indicts that peptide identification is performed by tandem mass spectra (MS/MS). Tandem mass pectrometry (MS/MS), however there is no information on how the sequence is determined. Is it by a database driven or de novo sequencing approach? The author should make mention of whether the fruit residue derived proteins been sequenced?
I also think that the authors should make mention of the use of food derived antioxidant proteins/protein hydrolysates/peptides as natural food preservation agents (e.g. prevention of lipid peroxidation etc).
Commercial proteolytic enzyme preparations (such as Alcalase and Flavourzyme) should have a capital letter as they are commercial products.
Line 18+88 and 27: Remove ‘s’ from ‘proteins’ and ‘proteomics’
Line 23: change ‘very usual’ to ‘the usual procedure’
Line 55: insert ‘of’ after ‘Regardless’
Line 70: replace ‘their’ with ‘to’ and ‘by’ with ‘on’
Line 130: replace ‘Ordinary’ with ‘traditional’
Line 144: replace ‘no-denaturing’ with ‘non-denaturing’
Lines 293-295: Rewrite the sentence as it doesn’t make sense
Author Response
Response to the reviewer #2.
Dear Editor,
Thank you very much for the corrections and suggestions regarding manuscript foods-842652. We have answered all your questions and we have made all corrections in the new version of the manuscript. An itemized explanation to your comments is below.
COMMENTS
*The authors need to provide comments and conclusions which they draw from their accumulation of evidence.
Following the reviewer suggestion, many comments, insights, and conclusions have been added all along the review.
*The authors need to indicate that some of the reagents are not food-grade (e.g., Urea, SDS, DTT, mercaptoethanol) and if the intended applications are for human/animal consumption then alternative extractants are required.
Thank you so much for the suggestion. The required information has been added accordingly.
*They need to comment on how applicable the methodology used is to a food industry environment.
Comments about industrial applicability has been added all along the manuscript (lines 144-145, 209-210, 252-255, 283-284, 335-337, 358-361, 390-392, 410-412, 418-420).
*There is limited information on the state of the raw material prior to protein extraction, is it wet/dry, is it milled etc, what is the best format for extraction and is this feasible on an industry basis.
Lines 133-139 have been added to describe the general procedure employed for material pre-treatment (drying, milling, defatting…) and why it is done. The specific state of the raw material in each case was already included in table 1, column nº 3.
*A paragraph describing the quantities, types and characteristics of proteins in the fruit residues (and potentially a SDS PAGE profile of the proteins) should be inserted in section 4 ‘Techniques used in the extraction and purification of proteins’ following paragraph 1 to provide the reader with basic background information before the author delves into the methods used for their extraction.
Thank you so much for your valuable suggestion. Nevertheless, it is not possible to generalize on the quantity, types, and characteristics of proteins in fruit residues. Every fruit residue has its own and characteristics proteins. However, a paragraph concerning the amount of proteins in fruit wastes has been added (lines 185-192) and characteristics of these proteins are described throughout the text.
*There is no information on how the sequence is determined. Is it by a database driven or de novo sequencing approach? The author should make mention of whether the fruit residue derived proteins been sequenced?
The requested information has been added to the manuscript (lines 1000-1005).
*The authors should make mention of the use of food derived antioxidant proteins/protein hydrolysates/peptides as natural food preservation agents (e.g. prevention of lipid peroxidation etc).
Thank you so much for your advice. The discussion about antioxidant proteins/protein hydrolysates/peptides applications has been further developed (lines 88-93).
*Orthographic suggestions (“Commercial proteolytic enzyme preparations (such as Alcalase and Flavourzyme) should have a capital letter as they are commercial products“, “Line 18+88 and 27: Remove ‘s’ from ‘proteins’ and ‘proteomics’”; “Line 23: change ‘very usual’ to ‘the usual procedure’”; “Line 55: insert ‘of’ after ‘Regardless’”,”Line 70: replace ‘their’ with ‘to’ and ‘by’ with ‘on’”,”Line 130: replace ‘Ordinary’ with ‘traditional’”, “Line 144: replace ‘no-denaturing’ with ‘non-denaturing’”, “Lines 293-295: Rewrite the sentence as it doesn’t make sense”)
Thank you very much for all your detailed suggestions. We have corrected the manuscript accordingly and language has been examined along the manuscript.

Reviewer 3 Report
In detail (for the Authors):
- a clear distinction between waste, production residue and by-product is needed;
- the reported antioxidant peptides obtained from fresh fruit could be improved including other fruits such as apple or grape (Mamma et al. 2009; Makris et al. 2007...);
-
Line 142 "90%": separate the value from the unity of measurement, please, be consistent in the whole manuscript;
- the potential application of antioxidant peptides should be explained more extensively.
Best regards.
Author Response
Response to the reviewer #3.
COMMENTS
*A clear distinction between waste, production residue and by-product is needed.
Thank you so much for your valuable suggestion. An explicit definition and distinction of each term has been added in lines 33-36.
*The reported antioxidant peptides obtained from fresh fruit could be improved including other fruits such as apple or grape (Mamma et al. 2009; Makris et al. 2007...).
There are three criteria that have been followed to select articles to be included in this review:
- The article should use fruit residues.
- There must be an extraction of peptides/proteins from the residue.
- The antioxidant properties of the peptides are assayed.
We have not been able to find any article fulfilling these criteria devoted to apple or grape wastes. The work of Mamma et al. (2009) was focused to the exploitation of solid by-products resulting from the processing of fruits (apple, orange, and grape) to obtain enzymes, bioethanol, organic acids, heteropolysaccharides, aroma compounds, protein enriched feeds, prebiotic oligosaccharides and biologically active molecules, but not bioactive peptides. The work of Makris et al. (2007) was devoted to the extraction and characterization of phenolic compounds, and not proteins or peptides, from grape wastes resulting during vinification.
*Line 142 "90%": separate the value from the unity of measurement, please, be consistent in the whole manuscript.
The mistake has been fixed. Thank you so much for the advice.
*The potential application of antioxidant peptides should be explained more extensively.
Thank you so much for your suggestion. The discussion about antioxidant proteins/protein hydrolysates/peptides applications has been further developed (lines 88-93).
Round 2
Reviewer 2 Report
The manuscript has improved significantly and warrants publication